# Could Phosphorous MR Spectroscopy Help Predict the Severity of Vasospasm? A Pilot Study

**DOI:** 10.3390/diagnostics14080841

**Published:** 2024-04-18

**Authors:** Malik Galijasevic, Ruth Steiger, Stephanie Alice Treichl, Wing Man Ho, Stephanie Mangesius, Valentin Ladenhauf, Johannes Deeg, Leonhard Gruber, Miar Ouaret, Milovan Regodic, Lukas Lenhart, Bettina Pfausler, Astrid Ellen Grams, Ondra Petr, Claudius Thomé, Elke Ruth Gizewski

**Affiliations:** 1Department of Radiology, Medical University of Innsbruck, 6020 Innsbruck, Austria; malik.galijasevic@i-med.ac.at (M.G.); stephanie.mangesius@i-med.ac.at (S.M.); valentin.ladenhauf@i-med.ac.at (V.L.); johannes.deeg@i-med.ac.at (J.D.); leonhard.gruber@i-med.ac.at (L.G.); miar.ouaret@i-med.ac.at (M.O.); mregodic@outlook.com (M.R.); lukas.lenhart@i-med.ac.at (L.L.); astrid.grams@i-med.ac.at (A.E.G.); elke.gizewski@i-med.ac.at (E.R.G.); 2Neuroimaging Research Core Facility, Medical University of Innsbruck, 6020 Innsbruck, Austria; 3Department of Neurosurgery, Medical University of Innsbruck, 6020 Innsbruck, Austria; stephanie.goerke@i-med.ac.at (S.A.T.); wing.ho@i-med.ac.at (W.M.H.); ondra.petr@i-med.ac.at (O.P.); claudius.thome@i-med.ac.at (C.T.); 4Department of Neurology, Medical University of Innsbruck, 6020 Innsbruck, Austria; b.pfausler@i-med.ac.at

**Keywords:** subarachnoidal hemorrhage, brain aneurysm, phosphorous spectroscopy, magnesium, cerebral pH, vasospasm, delayed ischemia

## Abstract

One of the main causes of the dismal prognosis in patients who survive the initial bleeding after aneurysmal subarachnoidal hemorrhage is the delayed cerebral ischaemia caused by vasospasm. Studies suggest that cerebral magnesium and pH may potentially play a role in the pathophysiology of this adverse event. Using phosphorous magnetic resonance spectrocopy (31P-MRS), we calculated the cerebral magnesium (Mg) and pH levels in 13 patients who suffered from aSAH. The values between the group that developed clinically significant vasospasm (*n* = 7) and the group that did not (*n* = 6) were compared. The results of this study show significantly lower cerebral Mg levels (*p* = 0.019) and higher pH levels (*p* < 0.001) in the cumulative group (all brain voxels together) in patients who developed clinically significant vasospasm. Further clinical studies on a larger group of carefully selected patients are needed in order to predict clinically significant vasospasm.

## 1. Introduction

More than three decades have passed since the pivotal International Cooperative Study on the Timing of Aneurysm Surgery [1], which conclusively demonstrated that vasospasm stands as the predominant cause of death in cases of aneurysmal subarachnoid hemorrhage (aSAH). Despite the strides made in understanding and managing this condition, complications arising from aSAH, and particularly delayed cerebral ischemia (DCI), persist as the primary sources of concern within the realm of the neurocritical care for patients who successfully survive the initial hemorrhage and are promptly transferred to specialized centers [2].

DCI remains the single-most significant contributor to both mortality and morbidity among individuals recovering from aSAH, underscoring the imperative need for comprehensive research and therapeutic interventions in this domain [3]. The multifaceted nature of DCI is underscored by a myriad of contributing factors, with vasospasm emerging as a central and pivotal element. In addition to vasospasm, other crucial determinants encompass microcirculatory dysfunction, glymphatic impairment, inflammatory responses, and neuroelectric disruptions [4].

As we delve into the intricate landscape of DCI, it becomes evident that a thorough understanding of its pathophysiology is paramount for advancing effective treatment strategies. The interplay among various factors underscores the complexity of DCI, urging researchers and clinicians alike to explore novel avenues for intervention and management. This expanded perspective not only acknowledges the historical context established by earlier studies but also highlights the ongoing challenges and opportunities for furthering our comprehension and, ultimately, improving outcomes for patients grappling with the aftermath of aSAH.

The ability to predict which patients will develop clinically significant vasospasm and its consequential complications could have a huge impact on the treatment and possible escalation of prevention strategies of vasospasm-related delayed cerebral ischaemia. This has been a point of research interest for some time. Various tests and predictors have been suggested in an attempt to predict which patients will develop clinically relevant vasospasm. For example, the Fisher scale and the modified Fisher scale use the computed-tomographic (CT) characteristics of the hemorrhage, like the amount of blood products, or the presence of intraventricular hemorrhage [5]. The modified Fisher scale has remained a significant predictor of clinically relevant vasospasm even after the correction for other known risk factors like early angiographic vasospasm, a history of hypertension, Hunt and Hess grades, and an elevated mean arterial pressure during admission [5].

The Monitoring and Imaging of SAH and Aneurysm (MISA) study is a prospective cohort trial in patients undergoing treatment for non-ruptured and ruptured aneurysms [6]. Besides the analysis of various biomarkers and CSF components, and besides standard neurocritical care, a phosphorous magnetic resonance spectroscopy (31P-MRS) was conducted after therapy in every patient.

Phosphorous magnetic resonance spectroscopy (31P-MRS) is an MRI spectroscopy technique that enables the in vivo and non-invasive measurements of energy metabolites and membrane turnover metabolites directly, and Mg and pH indirectly in a voxel of tissue.

An especially important factor in post-hemorrhagic vasospasm could be intracerebral magnesium (Mg). Hypomagnesemia occurs in approximately 50% of patients with aSAH. The addition of Mg to the therapy protocol showed a reduction in the stroke size and severity in aSAH stroke models [7,8,9]. Recent studies showed potentially reduced morbidity and mortality of aSAH patients after introducing magnesium sulfate therapy intravenously; however, most of these studies were inconclusive with mixed results [10,11,12,13]. Further studies suggested intrathecal magnesium sulphate injections, with somewhat better results [14,15]. Apart from nimodipine, no drugs showed a benefit in phase III trials, despite promising results in phase II trials. One of the reasons for the mixed results could be the observed incongruence in the designs of phase II and phase III trials [16]. One of the therapy regimens that showed enormous potential in phase II trials is the use of intravenous magnesium at a steady rate of 64 mmol/L/day [17]. However, the phase III trials did not confirm these results [18,19].

Another factor in this examined cohort could be brain tissue pH. pH, together with CO_2_, is a significant contributing factor to the cerebral blood flow [20].

In the pursuit of advancing our understanding of aSAH, the primary objective of this study was to undertake a comprehensive examination of cerebral magnesium (Mg) levels and cerebral pH in affected individuals. To achieve this, we employed advanced imaging techniques, specifically 31-phosphorus magnetic resonance spectroscopy (31P-MRS). This paper is an extension of a paper previously published by our group [6].

In this study, we quantified cerebral Mg levels and pH not only in the entire brain (cumulative results) but also with a focused analysis on the affected side of the brain. Additionally, we sought to delineate variations in these crucial metrics within the affected arterial territories, aiming to capture localized changes that may provide valuable insights into the pathophysiology of aSAH and DCI.

By employing 31P-MRS, we aimed to unravel the intricate interplay between cerebral Mg levels and pH, shedding light on potential correlations and patterns that could contribute to our understanding of the underlying mechanisms associated with aSAH. This comprehensive approach may contribute to possible targeted interventions and improved patient outcomes in this area.

## 2. Materials and Methods

Inclusion criteria for this study were aSAH, an MRI with a 31P-MRS sequence, and sufficient clinical and imaging follow-up data. The exclusion criteria were an age under 18 years, a hemorrhage due to other causes, or a repeated aneurysm rupture. Thirteen patients with aSAH who were admitted to our neurosurgery department between July 2016 and October 2017 were scanned using magnetic resonance imaging (MRI) in the first week after the hemorrhage (mean of 4 days after the event, always within 72 h from surgical or endovascular therapy). This corresponds to the time before the occurrence of clinically significant vasospasm, which develops at around 4–15 days after the initial bleeding [21]. Besides the standard sequences according to the institutional protocol, the 31P-MRS was conducted. The patients were hospitalized in the neurosurgical intensive care unit and closely monitored. They were then divided into a group with and a group without clinically relevant vasospasm (defined by well-established clinical criteria). Vasospasm was confirmed using transcranial doppler sonography (TCD) or angiography (Figure 1). The most important criterion defining the clinically relevant vasospasm was the development of delayed cerebral ischemia confirmed by the MRI, or the need to undergo a repeated conventional angiography and intra-arterial therapy (angiography-guided intra-arterial nimodipine application or percutaneous transluminal balloon angioplasty). We analyzed the cumulative results (all investigated voxels together), only voxels from the affected brain territory, and voxels from the affected side of the brain exclusively.

The study was approved by the local Ethics Committee (AN2016-0032 359/4.8), and completed in compliance with the Declaration of Helsinki. Informed consent was obtained from each patient or her/his legal guardian.

### 2.1. Magnetic Resonance Imaging

MRI scans were performed on a 3T MRI scanner (Verio, Siemens Medical AG, Erlangen, Germany), and for the 31P-MRS sequence, a double-tuned 1H/31P volume head coil (Rapid Biomedical, Würzburg, Germany) was employed.

The 3D 31P-MRS block was planned on a sagittaly oriented T2-weighted sequence, acquired in advance and covering the entire cerebrum (3D T2 space with a voxel size of 1.2 × 1.2 × 1.2 mm^3^, and the following settings: repetition time (TR) = 3000 ms, echo time (TE) = 412.0 ms, acquisition time (TA) = 2:50). Air-filled cavities and bone, fat, and boundary regions were omitted, in order to avoid voxel contamination, performed as described in detail by Grams et al. [22]. MR spectroscopy was executed via CSI (chemical shift imaging) in order to obtain spatially well-resolved spectra from different chemical compounds within the volume of interest (VOI), and the corresponding acquisition time (TA) was 10:44 min. The volume of interest (VOI) was acquired with an 8 × 8 × 8 mm matrix and a field of view of 240 × 240 × 200 mm^3^, resulting in a 30 × 30 mm^2^ voxel size and a slice thickness of 25 mm. The sequence was executed with a WALTZ 4 proton decoupling, a repetition time of 2000 ms (TR), an echo time of 2.3 ms (TE), and a flip angle of 60^∘^ (FA), and the number of averages was 10. Similar variations of this sequence have already been carried out in various studies [23,24,25,26,27,28,29,30].

Voxels that were included for the 31P-MRS analyses were chosen from the following vascular territories: the basilar artery (BA), anterior cerebral arteries (ACA), middle cerebral arteries (MCA), and the posterior cerebral arteries (PCA). Predetermined by the size of the measured voxels, we chose at least four voxels on each side of the brain within the regions of interest and averaged the calculated values for both hemispheres. Furthermore, every spectrum in each single voxel was visually assessed and checked regarding the quality of the spectra according to the criteria of existing literature [31]. Voxels that did not fulfill these quality criteria were excluded from further analysis.

Data post-processing of the 31P-MRS .rda data files (Siemens Medical AG, Erlangen, Germany) was performed offline with jMRUI (version 5.0, MRUI Consortium, available at http://www.mrui.uab.es, accessed 1 October 2020). The fitting process employed the non-linear square fitting algorithm AMARES and involved 12 Lorentzian-shaped exponentially decaying sinusoids, representing metabolites, including phosphocholine and phospho-ethanolamine (summarized as phsophomonoesters—PME), Pi, glycerophosphocholine, and glycerophosphoethanolamine (summarized as phosphodiesters—PDE), PCr, and ATP.

We calculated the pH and Mg^2+^ values as described in detail by Hattingen et al. [27]. The changes in pH were estimated from the signal position of inorganic phosphate and the PCr peak (δPi), and the changes in Mg^2+^ concentrations were assessed from the chemical shift difference between the β-ATP peak and the PCr signal (δβ). Both parameters were calculated according the formulae given by Iotti et al. [32], as described in detail in [27,33].



pMg=4.24−log10[(18.58+δβ)0.42/(−15.74−δβ)0.84]



pH=6.706−0.0307[Mg]+log10[(δPi−3.245)/(5.778−δPi)]



The analysis of the 31P-MRS data took place in all investigated voxels (cumulative results), and separately only in the affected side and in the arterial territory of the affected artery. The positioning of the voxels is shown in Figure 2.

### 2.2. Statistical Analysis

Statistical analysis was performed using R (R Core Team v. 3.6.1). The Shapiro–Wilk test and one-sample Kolmogorov–Smirnov test were used to assess the normality of the data, which was presented with quantile-comparison plots and histograms. The data were not normally distributed. The Wilcoxon signed-rank test was used for comparisons between groups. For the analysis of more than two groups, the Kruskal–Wallis test and Dunn’s post-hoc test were used. *p*-values < 0.05 were considered statistically significant. The results were presented using boxplots. Additionally, a sensitivity and specificity analysis was calculated and the results were presented using receiver operating characteristics curves. The cut-off values were determined using the greatest Youden’s J index. The statistics were performed using the cumulative results of all measurements, and using measurements from the affected side and arterial territory exclusively, as described in the Methods section.

## 3. Results

### 3.1. Patient Characteristics

This study incorporated a total of 13 patients, with a mean age of 55 years (±standard deviation 9.66 years), who had experienced aSAH. The characteristics of the participants are outlined in Table 1. More details concerning the patient group can be found in [6]. The cohort was a mixture of different clinical presentations, aneurysm locations, and applied treatments.

### 3.2. Cumulative Results

#### 3.2.1. Magnesium Levels

In individuals who experienced clinically relevant vasospasm, there was a notable and statistically significant reduction in cumulative cerebral magnesium levels across all investigated brain voxels (*p* = 0.019). This observation is visually represented in Figure 3.

This was especially pronounced in patients with anterior communicating artery (AComm) and anterior cerebral artery (ACA) aneurysms (*p* < 0.001) (Figure 4).

There was no statistically significant difference detected among patients with varying Hunt and Hess clinical grades, as illustrated in Figure 5.

#### 3.2.2. Cerebral pH Levels

Patients who developed clinically relevant vasosopasm had significantly higher cerebral pH levels compared to the patients who did not develop vasosopasm (*p* = 0.007) (Figure 6).

#### 3.2.3. Sensitivity and Specificity Analysis of Cerebral Mg and pH

The analysis for sensitivity and specificity derived an AUC value of 0.55 for Mg and 0.56 for pH (Figure 7).

### 3.3. Results from the Affected Arterial Territory and Affected Side

No significant difference could be derived from the Mg or pH levels when comparing only voxels in the affected side or affected arterial territory. There was no significant difference in the results when comparing the two different treatments arms (endovascular coiling and neurosurgical clipping).

## 4. Discussion

Aneurysmal subarachnoidal hemorrhage remains one of the most deadly neurological diseases with high morbidity and mortality rates. For those who survive the initial subarachnoidal bleeding and seek medical attention, vasospasm emerges as a predominant concern, with its complications, particularly the onset of DCI, being a leading cause of adverse outcomes. In the quest to enhance neurocritical care, predicting which patients are at risk of developing clinically significant vasospasm and implementing timely preventive measures has become an important goal.

Numerous studies have diligently explored various clinical tools to achieve this objective. Notably, Toi et al. employed transcranial Doppler ultrasound at different time points throughout the neurocritical care continuum following aSAH. Their findings revealed an elevation in the mean flow velocity (MFV) within the horizontal portion of the middle cerebral artery (MCA) in patients who later manifested clinically relevant vasospasm, showcasing the potential of this modality for early detection [34]. Building on this approach, Matamoros et al. expanded the scope to include bilateral anterior cerebral arteries (ACA) in their assessment. Their results echoed those of Toi et al., demonstrating an increased blood flow in patients predisposed to significant vasospasm [35]. Meanwhile, TCD became standard in the intensive care evaluation of aSAH patients. Souissi et al. used jugular bulb oximetry monitoring and found increased cerebral oxygen extractions (AVDO2) 12 h before the significant vasospasm occurred clinically [36]. Similar results were achieved in a study by Heran et al. [37].

The cumulative findings from these studies underscore the pivotal role of advanced imaging techniques and diagnostic tools in the early identification of vasospasm risk following aSAH. The pursuit of predictive strategies not only promises potential improvements in patient outcomes but also signifies a critical advancement in the ongoing evolution of neurocritical care. By proactively identifying and addressing the risk of vasospasm, these studies contribute to a more comprehensive and personalized approach to managing individuals grappling with the consequences of aSAH. This forward-looking perspective holds the potential to enhance both the understanding and the treatment of this complex neurological condition, ultimately paving the way for more effective and tailored interventions in the field of neurocritical care.

Some clinical predictors can also be used in order to predict clinically relevant vasospasm. Csok et al. showed very good results using the Hijdra sum score, early sonographic vasospasm, and a simplified binary level of the consciousness score between admission and day 5 after the event [38]. Various studies demonstrated hypomagnesemia at some time point following aSAH [39,40].

Previous studies examined the role of cerebral Mg and pH in the patophysiology of vasospasm after aSAH. To our knowledge, this is the first study using 31P-MRS in the assessment of Mg and pH in patients with SAH and clinically relevant vasospasm. The results of this study show the potential of 31P-MRS for indicating the presence of clinically relevant vasospasm in the first few days after aSAH, using Mg and pH. Our results show decreased Mg and increased pH levels in patients that went on to develop clinically relevant vasospasm. This was especially true for the aneurysms of the AComm and ACA. This correlation was not associated with the Hunt and Hess scores in clinical presentation. Even though the results of this study show some significant difference in the cerebral Mg and pH in patients who developed clinically relevant vasospasm, the sensitivity and specificity analysis showed no statistical significance. We also undertook the analysis of the results using only voxels from the affected arterial territory and only voxels from the side of the bleeding. These results show no significant difference between the patients with and patients without clinically relevant vasospasm in voxels positioned in bleeding areas. This could be due to a number of reasons. We hypothesize that the main reason for this is the change of the micro-environment in these regions due to the oedema and blood products. In our patient group, there were no differences between the two treatment arms (clipping vs. coiling).

Our results regarding Mg are in agreement with other studies, albeit with different methods. Wipplinger et al. showed reduced serum Mg concentrations in patients with vasospasm after aSAH and the importance of magnesium sulfate administration with a careful titration to 2–2.5 mmol/L of serum magnesium [41]. Another study showed an improvement in vasospasm markers (such as the cerebral blood flow) after the intraoperative administration of magnesium sulfate in patients after aSAH [42].

On the other hand, our pH results were somewhat contrary to the results of other studies. For example, Charbel et al. showed significantly lower brain tissue pH levels in patients experiencing vasospasm using invasive monitoring [43]. Other studies found higher CSF pH levels to be a contributing factor in the development of delayed cerebral ischaemia [44]. It is worth noting that these studies used different methods and, partly, different patient collectives.

The potential utility of 31P-MRS in indicating the presence of clinically significant vasospasm is highly promising, especially considering its non-invasive nature. This method could offer valuable insights into the physiological changes associated with vasospasm after SAH.

However, it is crucial to acknowledge that the precise pathophysiological mechanisms underlying the alterations in pH and Mg levels following aSAH are not yet fully understood. The complex interplay of various factors in this context necessitates further exploration to unravel the intricate details of these changes. Moreover, it is essential to recognize certain limitations of the present study. The size and positioning of the 31P-MRS voxels pose challenges, as it may not always be feasible to position a voxel exclusively within the affected arterial territory. The inherent size constraints could lead to partial volume effects, potentially contaminating some of the voxels and influencing the accuracy of the measurements. Also, it would be interesting to see the differences between various grades of vasospasm. As our patient cohort was not big enough for this analysis, we omitted it from this publication. In future studies, we plan to test a vasospasm grading system, like the CTA vasospasm score developed by van der Harst et al. [45], or the grading on digital subtraction angiography developed by Merkel et al. [46]. Also, the longitudinal analysis of the data would also be of interest. As the intensity of vasospasm peaks at around 4–15 days after bleeding [21], and our imaging occurred at the mean of 4 days after the bleeding event, this could also have influenced the results. Furthermore, our cohort included a small number of patients with different clinical presentations and radiographic features, and two different treatment arms. A carefully planned prospective study including only patients with anterior circulation aneurysms and only one treatment arm, together with a standardized positioning of the 31P-MRS voxels, is needed.

These considerations emphasize the need for the continued refinement and optimization of imaging techniques to overcome such limitations and enhance the reliability of 31P-MRS as a diagnostic tool in the assessment of vasospasm after aSAH. Also, new technologies like AI, data-driven analysis, and automated classification will probably be needed. For example, Roederer et al. showed the possibility of the automated classification of the vasospasm class in patients after aSAH [47]. Our results show slight significant differences in Mg and pH measurements in the brain, albeit with strong overlaps. We hypothesize that adding 31P-MRS to a plethora of clinical and imaging findings could, with the help of advanced tools, predict the severity of vasospasm.

## 5. Conclusions

This pilot study has illuminated the potential of employing magnesium (Mg) and pH measurements through 31P-MRS in the context of aSAH. The findings reveal a discernible trend, indicating lower magnesium levels and higher pH levels in patients who subsequently developed clinically relevant vasospasm. However, it is crucial to acknowledge that the study’s conclusions are constrained by several factors. Despite the limitations, the study suggests a promising option for further exploration. The measured Mg and pH levels could potentially serve as valuable additions to other diagnostic tests, contributing to a more personalized approach in anti-vasospasm therapy for patients after aSAH. To solidify these preliminary findings and establish their clinical relevance, it is imperative to conduct larger-scale prospective clinical studies. A standardization of measurements is needed for a better comparison between patient groups. These studies should adhere to strictly defined imaging and clinical criteria to identify the most promising and feasible tests for predicting clinically relevant vasospasm and informing the treatment strategies for vasospasm and its associated complications. This iterative process will ultimately enhance our understanding and management of aSAH, potentially leading to improved outcomes and individualized care for affected patients.

## Figures and Tables

**Figure 1 diagnostics-14-00841-f001:**
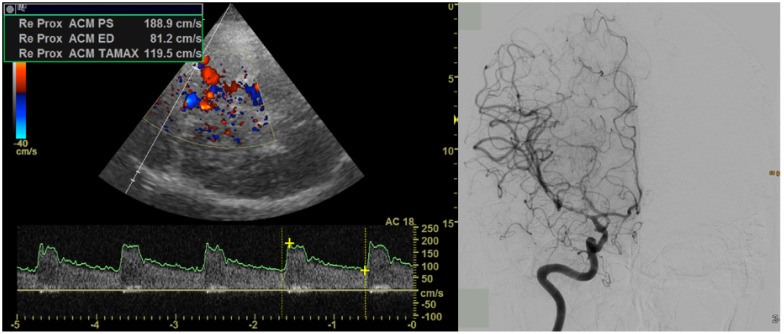
Vasospasm was confirmed using TCD (**left**, in a patient with a distal right-sided M1-vasospasm and a mean flow velocity of 120 cm/s) or angiography (**right**, in another patient with right-sided M1-vasospasm.

**Figure 2 diagnostics-14-00841-f002:**
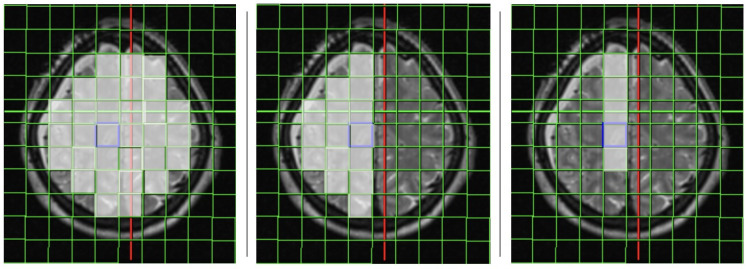
Positioning of the 31P-MRS voxels in a patient with a ruptured right-side ACA-aneurysm: all voxels (**left**), only the affected side (**middle**), and only the affected territory (**right**).

**Figure 3 diagnostics-14-00841-f003:**
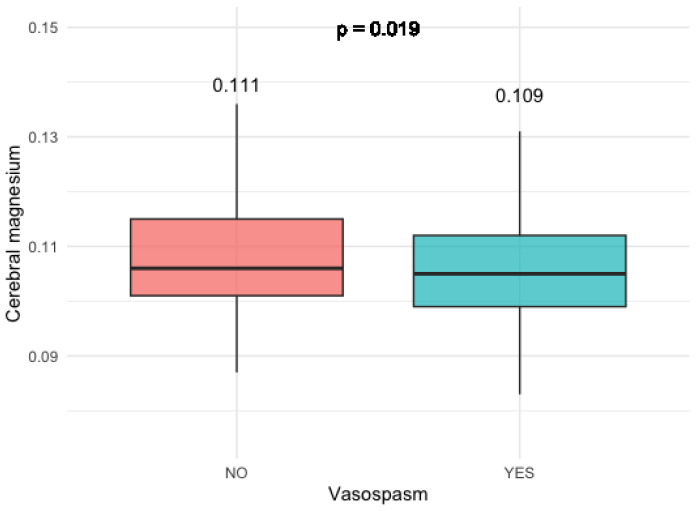
Magnesium levels in patients with and without relevant vasospasm.

**Figure 4 diagnostics-14-00841-f004:**
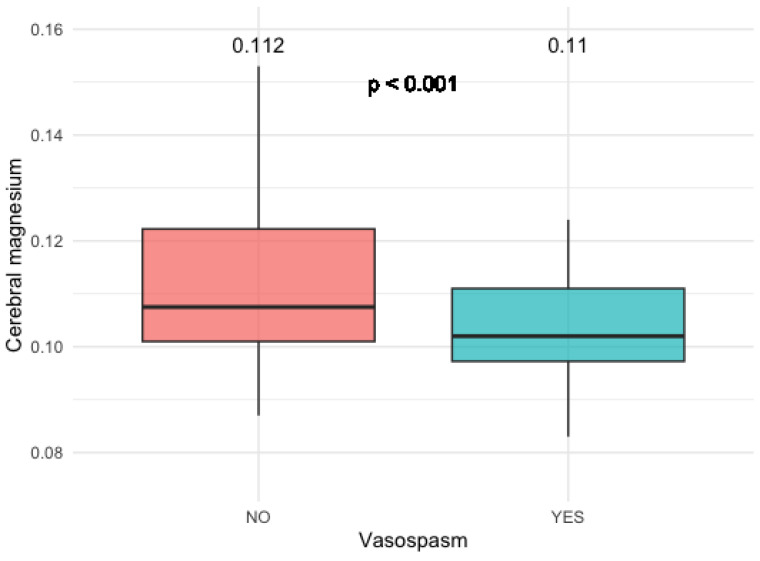
Magnesium levels in patients with SAH from AComm and ACA aneurysms.

**Figure 5 diagnostics-14-00841-f005:**
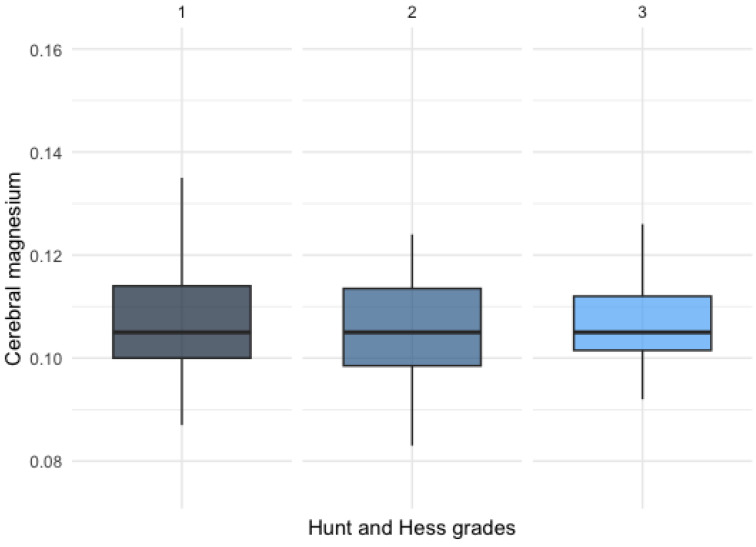
Magnesium levels in patients with different Hunt and Hess grades.

**Figure 6 diagnostics-14-00841-f006:**
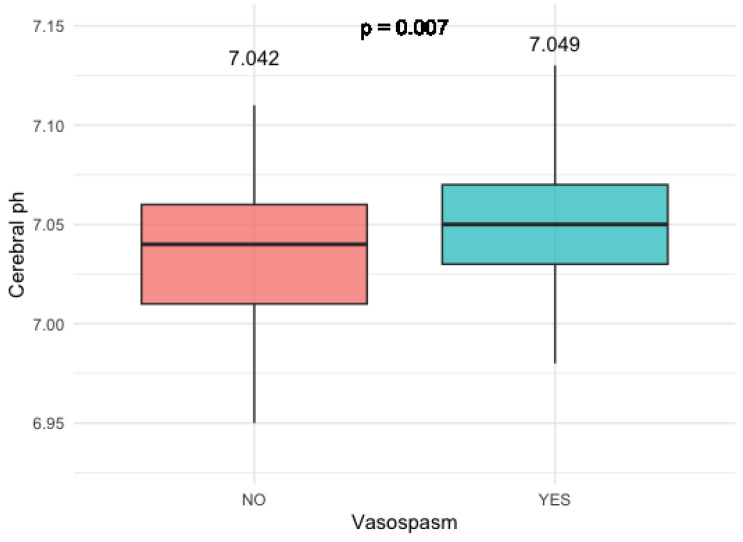
pH levels in patients with and without relevant vasospasm.

**Figure 7 diagnostics-14-00841-f007:**
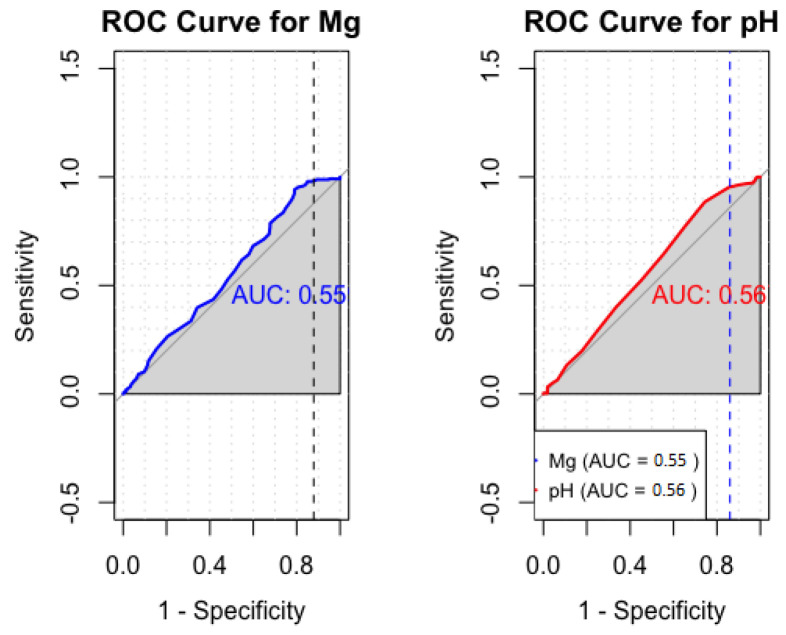
Sensitivity and specificity analysis of cumulative cerebral Mg and pH. The Youden index is represented by a vertical dotted line.

**Table 1 diagnostics-14-00841-t001:** Patients’ characteristics.

Study Participants	n (%)
Number of patients	13
Female	11 (84.6)
Male	2 (15.4)
Mean age	55 ± 9.6
Applied treatment	
Clipping	9 (69.2)
Coiling	4 (30.8)
Aneurysm location	
ICA	1 (7.6)
MCA	2 (15.4)
AComm	5 (38.4)
ACA	2 (15.4)
PComm	3 (23.1)
Patients with vasospasm	7 (53.8)

## Data Availability

The research data will be made available upon request to the corresponding author.

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
