# Peer review of "Could Phosphorous MR Spectroscopy Help Predict the Severity of Vasospasm? A Pilot Study"

_diagnostics, 2024, doi:10.3390/diagnostics14080841_

Round 1
Reviewer 1 Report
Comments and Suggestions for Authors
This is a well written paper on the impact of cerebral Mg2+ levels and of pH-value on vasospasm after subarachnoid hemorrhage. The topic is of paramount importance in the treatment of SAH and the authors demonstrate that measurement of Mg2+ and of pH by specialized MRI-techniques in a clinical setting is possible. On the other hand, the results are disappointing: Although there is a significant difference between both groups (with and without vasospasm) the overlap is large, and the AUCs are bad, so that the data presented in this paper does not support the use of Mg2+ and of pH as an aid for clinical decision making in the individual patient.
Minor points:
Please provide a clearcut definition of vasospasm.
Please show the correlation between Mg2+ and pH with the results of transcranial Doppler sonography
Author Response
Dear reviewer,
thank you very much on you constructive comments which made our manuscript better.
We provided a definition of vasospasm in our case (classical symptoms and imaging confirmation). The exact correlation of TCD and spectrosopy results is not possible, because the vasospasm was sometimes confirmed using conventional angiography, and not all patients recieved TCD at the time of the MRS. However, this is an interesting point and we will include it in the planning of our next research. The examples of TCD and conventional angiography are now given in the Figure 2.
Best regards
Reviewer 2 Report
Comments and Suggestions for Authors
Dear authors, I had the opportunity to read your interesting manuscript.
Major recommendations for a revision:
-please clearly state in all parts of the manuscript, that the current paper is an extension of the 31p-MRS data analyses in Treichl et al Front Neurol 2022 as already correctly indicated by you in connection with table 1 (ref 7). This is not a flaw but for the reader it is more understandable, why no PCr/APT etc. ratios are reported here (as this were reported already in 2022).
-currently, the Discussion is quite tedious to read; repetitions such as „to our knowledge this is the…“ at line 225/226 and again at 254/256; limitations twofold and incongruent positioned; hints to further studies at various sites; please re-write the Discussion, starting with the Main findings, then proceding with subheadings discussing the both items Mg and afterwards pH, then summarizing the limitations and omiting the clinical relevance, which should be positioned in the conclusion.
Minor recommendations for a revision:
-methods: please include TA 31P-MRS; is it a CSI method?
-results: fig 3 - in the text p<0.05, imbedded in the fig p<0.001; text doubling prior fig 4 (Hunt and Hess); inconsistencies for AUC numbers in fig 6 (Mg in the left part in blue 0.55 versus 0.56 in the text lines in the right part; the same for pH - 0.56 versus 0.57; in the text the numbers of red and blue are used)
-discussion: typo in line 196/197 imAportant;-); Fisher scale „pops-up“ in lines 240/241 ff - please show numbers in methods and results and then discuss, or omit;
Kind regards
Comments on the Quality of English LanguageMinor revision necessary- especially for the Discussion.
Author Response
Dear reviewer, thank you very much on you constructive comments which made our manuscript better.
- We now clearly stated in the text that this paper is an extension of a paper previously published by our group (Treichl et al.).
- The discussion is now re-written as per your recommendation. We did not highlighted the new sentences because of the extensive changes.
- The 31P-MRS methods are extended to include TA and explanation that it is a CSI method.
- The results are now consistent with the figures.
- Fisher scale is now omitted from the manuscript as there was no statistical significance with both the clinical measurements nor with the MRS.
Best regards